# Analysis of Strategy Selection in Third-Party Governance of Rural Environmental Pollution

**Qianwen Wu** [1,2], **Qiangqiang Wang** [1,2] and **Yongwu Dai** [1,2,*]

1   School of Economics and Management, Fujian Agriculture and Forestry University, Fuzhou 350002, China;
    cheerfulwqw@163.com (Q.W.); fjwqqiang@163.com (Q.W.)
2   Enterprise Management Research Center, Fujian Agriculture and Forestry University, Fuzhou 350002, China
*   Correspondence: fjdyw@163.com

**Abstract:** In the context of increasingly prominent rural environmental problems, the third-party pollution governance model has become an important initiative for the comprehensive governance of rural environmental problems in China. However, the current third-party governance initiatives continue to suffer from governance failures caused by unclear responsibility sharing and opportunistic behavior. To analyze the reasons behind the behavioral choices of the various stakeholders involved in rural environmental third-party governance and to provide policy insights for formulating more reasonable rural environmental third-party governance solutions, a game model was constructed between local governments and third-party governance institutions. Specifically, the model examined the different evolutionary game strategies that appear between local governments and third-party governance institutions in different institutional design contexts when responsibility disputes arise in third-party governance. These disputes involve the re-governing of pollution control responsibility, which can be borne either by the local governments or the party causing the damage. The results shown are as follows: It is crucial to define the boundaries of re-governance responsibility in the third-party governance of rural environmental pollution. When local governments bear the primary responsibility for governance, regardless of whether they provide regulatory oversight, third-party governance institutions tend to adopt a passive approach. In such cases, the third-party governance market fails to effectively fulfill its role in governance. By reconstructing the third-party governance market model and dividing the main responsibility for pollution governance among the damaging parties, it is possible to achieve active governance by third-party governance institutions without the need for regulation by local governments.

**Keywords:** negative governance; responsibility boundaries; regulation; evolution game; Jacobian matrix

## 1. Introduction

Since entering the 21st century, China's rural environmental problems have become increasingly prominent. Due to inadequate environmental infrastructure, domestic waste fails to be effectively disposed of in a timely fashion, leading to a common environmental problem: the widespread dumping of domestic sewage and solid waste [1,2]. In response to the significant problems faced by the rural environment, China's No. 1 Central Document of 2018, "Laying down Opinions of the Central Committee of Communist Party of China and the State Council on Implementing the Strategy of Rural Vitalization", points out that a pleasant living environment is the key to rural vitalization, a good ecological environment is the greatest advantage and most valuable asset of rural areas, and the comprehensive governance of rural environmental problems should be strengthened [3]. Under the traditional rural environmental governance model, local governments, as single governance bodies, have many limitations and deficiencies related to technology and regulation [4–6], which restrict the efficiency of environmental governance and make it difficult to effectively control the rural environmental surface source pollution situation [7,8].

　　　Scholars from various countries around the world have conducted extensive research on these issues. Hassan et al. (2020) argued that good governance contributes to reducing rural pollution levels [9]. Bildirici (2022) utilized panel data to test the impact of governmental governance on the environment in certain countries in the Middle East and sub-Saharan Africa, and found that poor governance models led to continued deterioration in pollution [10]. Amaruzaman et al. (2022) demonstrated the significance of local governmental governance in addressing environmental pollution through the application of a polycentric governance approach [11]. Pargal et al. (1996) confirmed the feasibility of introducing third-party governance institutions to control environmental pollution in Indonesia [12]. Raynolds et al. (2007) proposed that effective environmental governance in North America requires not only public regulation by government entities, but also active cooperation from the private sector [13]. Additionally, Driessen et al. (2012) proposed a framework by applying it to two environmental policy sectors in the Netherlands, and further demonstrated the importance of the governance capacity of local governments [14]. Van der Kamp et al. (2017) argued that China's traditional model of environmental pollution governance has primarily followed a government-led, top-down regulatory approach to the control of pollution behavior. This governance model has not only increased the regulatory costs for local governments, but, more importantly, has also failed to effectively reduce the levels of environmental pollution [15].

　　　Due to changes in government functions and the enrichment of environmental pollution control methods, the market is now playing an increasingly important role in environmental pollution control [16,17]. China proposed third-party environmental pollution control in 2013 for the first time [18,19]. In this new mode of environmental pollution control, polluters pay environmental service enterprises to conduct pollution treatment according to a contract [20] similar to an environmental service contract (ESC). To date, much progress has been made in third-party environmental pollution control in China [21–23]. By introducing third-party governance into rural environmental pollution governance, local governments can transfer the task of pollution governance to third-party governance institutions after paying certain fees and by signing contracts or agreements with them. Meanwhile, third-party governance institutions, as professional environmental service enterprises, can provide more complete pollution governance facilities, standardized pollution governance modes, and professional technologies, which not only reduce the total cost of rural pollution governance, but also greatly improve the efficiency of rural environmental governance [24,25]. The emergence of the third-party governance model has gradually achieved the transition from "who pollutes, who governs" to "who makes pollution, who makes payments, and specializes in governance" [26], but the transition of the model has produced new impacts on the existing environmental pollution control field. First, there is the problem of opportunistic behavior caused by information asymmetry [27,28]. According to the reform of government functions, environmental pollution governance enterprises can enter the third-party governance market without approval [29]. Therefore, in the absence of a sound access mechanism and a perfect regulatory mechanism, the third-party pollution governance market services vary considerably in quality [30,31]. As local governments have difficulty obtaining complete and accurate pollution data, there may be no fault, but they must bear the responsibility for the poor governance of third-party governance institutions. For example, some third-party governance institutions may exploit the loopholes created by a local government's unfamiliarity with pollution governance to add exemption clauses to environmental service contracts to avoid the impacts of ineffective governance caused by negative governance. Therefore, local governments may face greater market risks in the process of selecting third-party governance institutions. In the event that third-party governance fails to meet the requirements of environmental regulation, local governments will lose the incentive to trust third-party governance and choose to insist on their own model of pollution governance, resulting in the third-party governance model being unable to play a professional and market-oriented role in rural environments [32]. Second, the unclear standard of responsibility sharing can easily lead to unfairness in the

third-party governance market [33,34]. In the traditional dual-structure relationship, the government is the main body responsible for environmental governance; local governments are responsible for the environments in their respective jurisdictions, and the responsibility for pollution governance can be clearly divided among local governments [35]. However, under the triadic structure theory, after the dischargers transfer the pollution governance responsibility to third-party pollution governance subjects, the previously simple responsibility pattern is broken. Under the third-party governance model, third-party governance institutions have a natural motive to pursue profits. They weigh the benefits of governance against the costs, aiming to achieve optimal returns while evading environmental regulations. This increases the challenges faced by governmental enforcement entities. Some scholars believe that pollution governance responsibility should be transferred to third-party governance institutions with the signing of the contract; that third-party governance institutions need to carry out governance according to the contracted quantity, type, concentration, and other emission conditions [25,36]; and that third-party pollution governance subjects should be held responsible as independent pollution governance subjects if there is unqualified governance. Others support the idea that third-party pollution governance subjects only need to undertake responsibility for governance according to the contract; that third-party governance should transfer the pollution control behavior to the third-party governance institutions, but not the legal responsibility [37]; and that they do not need to bear the responsibility for any re-treatment arising from ineffective governance [38]. The main responsibility for re-governance should be borne by local governments, regardless of which party is at fault for ineffective third-party governance [39], and if local governments can prove the fault responsibility of third-party governance institutions, those institutions can be held liable according to the contractual agreement. Other scholars advocate that local governments and third-party governance institutions should share joint and several responsibilities; if the level of responsibility can be determined, each should bear the corresponding responsibility; if it is difficult to determine the level of responsibility, they should be equally responsible, etc. [40,41].

In the face of complex and changing conflicts of interest among responsible parties, there is still no reasonable answer to the question of how local governments can play their regulatory and coordinating roles fairly and effectively to promote the healthy and orderly development of the third-party governance market for rural environmental pollution. The previous literature has mostly studied the dilemma of third-party governance from the perspectives of contract design and the coordination of third-party governance relationships, while the strategy of local governments has been mostly examined in preliminary strategy analyses of whether third-party governance is chosen, lacking the perspective of responsibility taking and the allocation of responsibility between local governments and third-party governance institutions when third-party governance is not effective [32]. In the context of unclear responsibilities and mutual shirking between local governments and third-party governance institutions, it is important to study whether local governments should regulate third-party pollution governance institutions to ensure the effective operation of the third-party governance of rural environments and to promote rural revitalization. In recent years, the evolutionary game model has been increasingly applied to environmental pollution management problems [42,43]. Most studies have involved environmental pollution games between governments and polluters [44,45]. Others have investigated whether games among regions or states could lead to cooperation in treating environmental pollution [46,47]. The results of all these studies have demonstrated that evolutionary game models can be used to effectively explore the issues related to environmental pollution governance. Therefore, this study constructed an evolutionary game model between local governments and third-party governance institutions, and simulated different evolutionary outcomes using MATLAB software. Specifically, it examined the stable strategies of both parties' evolution when different stakeholders assume the responsibility for re-governance, aiming to find a balance point among environmental regulation, liability definition, and en-

vironmental infringement in the third-party governance market, to provide novel insights for rural environmental pollution governance.

## 2. Materials and Methods

### 2.1. Method Choice

An evolutionary game model can help multiple interest groups to attain their own optimal strategies by continuously changing and updating through observation, learning, and imitation in accordance with the concept of maximum interest under the circumstance of limited rationality [48]. Evolutionary game theory possesses the following characteristics: it focuses on a specific group that undergoes changes over time, aiming to comprehend the dynamic process of group evolution and elucidate the reasons behind the group reaching a particular state and how such a state can be achieved. The factors that influence group change encompass both random and disruptive events as well as regular patterns that emerge through selection mechanisms during the evolution process. The majority of evolutionary game theories derive their predictive or explanatory capabilities from the selection processes of groups, which typically exhibit a degree of inertia. Concurrently, these processes also encompass the potential for mutation, consistently giving rise to new variations or traits [49].

After local governments entrust third-party governance institutions to carry out rural environmental management, they cannot fully grasp whether the third-party governance institutions will actively manage the rural environment in the future, and third-party governance institutions cannot fully grasp whether local governments will regulate the rural environmental management process. However, if the participants are good learners, each could attempt to maximize their interests by constantly imitating other players' strategies. Therefore, the evolutionary game model can effectively solve the problem of behavioral strategy selection with regard to rural environmental management.

### 2.2. Basic Assumptions

The law has not yet clarified the boundary of responsibility for the third-party governance of pollution [50], and it is difficult to identify local governments and third-party governance institutions when pollution governance problems arise. However, with the signing of third-party governance contracts, local governments can largely optimize their functions in environmental governance, transforming from being the main body of pollution governance to being the main body of regulation, and can regulate the behavior of third-party governance institutions who compromise project quality by saving costs and increasing profit levels [51]. When pollution governance is not effective and local governments cannot easily prove the fault of third-party governance institutions, local governments will bear the responsibility for re-governing pollution. Meanwhile, local governments can provide evidence of ineffective governance by third parties through regulation, and can obtain certain amounts of liquidated damages from third-party governance institutions to compensate for the losses caused by ineffective pollution governance.

**Assumption 1.** *Local governments may regulate whether third-party governance institutions meet the standards of pollution control according to the contract. Alternatively, some local governments think that, with the signing of the third-party governance contract, the main responsibility for pollution control has been transferred to the third-party governance institutions, so they do not expend energy on follow-up and regulation. Therefore, the behavioral strategies of local governments can be divided into regulation and non-regulation.*

**Assumption 2.** *Third-party governance institutions, as "rational economic agents," use improper measures to reduce costs in cases where the boundaries of responsibility are unclear, and when local governments do not regulate third-party governance institutions, those institutions may adopt negative governance approaches, such as formalities and reducing pollution governance inputs, to reduce costs. Alternatively, they may choose active governance because the local governments regulate the pollution governance process and they are required by those local governments to bear*

*high default costs when the governance is not effective. Thus, the behavioral strategy choices of third-party governance institutions can be classified as active or negative governance.*

**Assumption 3.** *This work assumes the local government and third-party governance institution strategies as follows: the local government behavioral strategy set is {regulation, non-regulation}; the third-party governance institution behavioral strategy set is {active governance, negative governance}. The probability of local governments choosing the "regulation" strategy is $x(0 < x < 1)$ and the probability of local governments choosing the "non-regulation" strategy is $1 - x$; the probability of third-party governance institutions adopting the "active governance" strategy is $y(y < 1 < 0)$ and the probability of third-party governance institutions choosing the "negative governance" strategy is $1 - y$.*

**Assumption 4.** *Assuming that local governments commission third-party governance institutions to carry out pollution governance, the costs of regulating the third-party governance institutions are represented by $C_J$. When third-party governance institutions fail to carry out pollution governance in accordance with the requirements, this results in local governments having to reassume environmental pollution governance due to unclear responsibilities, which is represented by $C_T (C_J > C_T)$. Third-party governance institutions undertaking pollution governance tasks and obtaining revenue is denoted by $R_Z$, the costs required for active governance are shown as $G_A$, and the costs required for negative governance are represented by $G_B (G_A > G_B)$. When local governments regulate and third-party governance institutions practice negative governance, third-party governance institutions are required to provide compensation for breach of contract, depicted as $R_J$.*

The related symbols and definitions are further described in Table 1.

**Table 1.** Description of parameter symbols.

| Parameter | Definition |
| --- | --- |
| $C_J$ | Regulatory costs when local governments commission third-party governance institutions to carry out pollution governance |
| $C_T$ | Governance loss caused by local governments reassuming responsibility for environmental pollution governance when third-party governance institutions fail to control pollution in accordance with the requirements |
| $G_A$ | Governance costs when third-party governance institutions choose the "active governance" strategy |
| $G_B$ | Governance costs when third-party governance institutions choose the "negative governance" strategy |
| $G_T$ | Re-governance costs that third-party governance institutions choose the "negative governance" strategy if the damaging parties are responsible for the re-governance of pollution |
| $R_Z$ | Benefits when third-party governance institutions undertake the task of pollution governance |
| $R_J$ | Compensation fee paid by third-party governance institutions to local governments when negative governance by third-party governance institutions is discovered |

### 2.3. Model Building

Based on the above assumptions, the payoff matrix of the two-player game between third-party governance institutions and local governments can be constructed as shown in Table 2.

**Table 2.** Payoff matrix of the game in rural environmental pollution governance.

| Game Participants | | Third-Party Governance Institutions | |
|---|---|---|---|
| | | **Active Governance** **(y)** | **Negative Governance** **(1 − y)** |
| Local governments | Regulation $(x)$ | $(-C_J, R_Z - G_A)$ | $(-C_J - C_T + R_J, R_Z - G_B - R_J)$ |
| | Non-regulation $(1 - x)$ | $(0, R_Z - G_A)$ | $(-C_T, R_Z - G_B)$ |

Note: The relationships of all formulas in the table are derived from the above assumptions.

### 2.3.1. Expected Benefits of Local Governments

Assuming that the expected return to local governments for adopting the "regulation" strategy is $E_{11}$, the expected return for adopting the "non-regulation" strategy is $E_{12}$, and the expected average return for adopting a mixed strategy is $E_1$, then $E_{11}$, $E_{12}$, and $E_1$ are expressed as in Equations (1)–(3), respectively:

$$E_{11} = y(-C_J) + (1-y)(-C_J - C_T + R_J) \tag{1}$$

$$E_{12} = (1-y)(-C_T) \tag{2}$$

$$\overline{E_1} = xE_{11} + (1-x)E_{12} \tag{3}$$

From Equations (1)–(3), the replication dynamics equation for local governments can be obtained as shown in Equation (4):

$$\mathrm{F}(x) = \frac{dx}{dt} = x(E_{11} - \overline{E_1}) = x(1-x)(E_{11} - E_{12}) = x(1-x)(R_J - C_J - yR_J) \tag{4}$$

The derivative of Equation (5) is obtained by taking the derivative of Equation (4):

$$\frac{dF(x)}{dx} = (1 - 2x)(R_J - C_J - yR_J) \tag{5}$$

where $\frac{dF(x)}{dx} = 0$ and $y^* = \frac{R_J - C_J}{R_J}$. When $y = y^* = \frac{R_J - C_J}{R_J}$, the expected return to local governments is maximized. When $y > \frac{R_J - C_J}{R_J}$, $\frac{dF(x)}{dx}\big|_{y=0} < 0$, according to the differential equation stability theorem, the evolutionary stabilization strategy for local governments is not to regulate. When $y < \frac{R_J - C_J}{R_J}$ and $\frac{dF(x)}{dx}\big|_{y=0} < 0$, the evolutionary stabilization strategy for local governments is to regulate. It can be seen that, if the probability of active governance by third-party governance institutions is greater than $y^*$, local governments will tend to adopt the "non-regulation" strategy; conversely, if the probability of active governance by third-party governance institutions is less than $y^*$, local governments will tend to adopt the "regulation" strategy.

### 2.3.2. Expected Benefits of Third-Party Governance Institutions

Assuming that the expected return to third-party governance institutions for adopting the "active governance" strategy is $E_{21}$, the expected return for adopting the "negative governance" strategy is $E_{22}$, and the expected average return for adopting a mixed strategy is $E_2$, then $E_{21}$, $E_{22}$, and $E_2$ are expressed as in Equations (6)–(8), respectively:

$$E_{21} = x(R_Z - G_A) + (1-x)(R_Z - G_A) \tag{6}$$

$$E_{22} = x(R_Z - G_B - R_J) + (1-x)(R_Z - G_B) \tag{7}$$

$$\overline{E_2} = yE_{21} + (1-y)E_{22} \tag{8}$$

From Equations (6)–(8), the replication dynamics equation for the third-party governance institutions can be obtained as shown in Equation (9):

$$F(y) = \frac{dy}{dt} = y(E_{21} - \overline{E_2}) = y(1-y)(E_{21} - E_{22}) = y(1-y)(xR_J + G_B - G_A) \quad (9)$$

The derivative of Equation (10) is obtained by taking the derivative of Equation (9):

$$\frac{dF(y)}{dy} = (1-2y)(xR_J + G_B - G_A) \quad (10)$$

where $\frac{dF(y)}{dy} = 0$ and $x^* = \frac{G_A - G_B}{R_J}$. When $x = x^* = \frac{G_A - G_B}{R_J}$, the expected return to third-party governance institutions is maximized. When $x > \frac{G_A - G_B}{R_J}$, $\frac{dF(x)}{dx}\big|_{x=1} < 0$, according to t the differential equation stability theorem, the evolutionary stabilization strategy for third-party governance institutions is active governance; when $x < \frac{G_A - G_B}{R_J}$, $\frac{dF(x)}{dx}\big|_{x=0} < 0$, the evolutionary stabilization strategy for third-party governance institutions is negative governance. It can be seen that, if the probability of regulation by local governments is greater than $x^*$, third-party governance institutions will tend to adopt the "active governance" strategy; conversely, if the probability of regulation by local governments is less than $x^*$, third-party governance institutions will tend to adopt the "negative governance" strategy.

## 3. Analysis of the Evolutionary Stable Strategies

When local governments and third-party governance institutions reach the maximum value at the same time, the strategy that they maintain will be the evolutionary stable strategy (ESS). When the determinant det ($J$) of the Jacobi matrix is positive and the trace tr ($J$) is negative, the equilibrium point of both games can be judged to be in a stable state. Otherwise, the point will be considered a saddle.

Solving the system of equations consisting of Equations (4) and (9), we can obtain four partial equilibrium points in the game of the processes of local governments and third-party governance institutions: (0, 0), (0, 1), (1, 0), and (1, 1).

### 3.1. The Evolutionary Stable Strategy for a Mixed Game When Local Governments Bear the Responsibility for Renewed Pollution Governance

The expression of the Jacobi matrix determinant when local governments assume responsibility for the re-governance of pollution and the expression of the trace can be obtained by systematically deriving the following system of differential equations [52]:

$$J_1 = \begin{bmatrix} \frac{dF(x)}{dx} & \frac{dF(x)}{dy} \\ \frac{dF(y)}{dx} & \frac{dF(y)}{dy} \end{bmatrix} \quad (11)$$

$$det J_1 = (1-2x)(R_J - C_J - yR_J)(1-2y)(xR_J + G_B - G_A) - x(1-x)R_J y(1-y)R_J \quad (12)$$

$$tr J_1 = (1-2x)(R_J - C_J - yR_J) + (1-2y)(xR_J + G_B - G_A) \quad (13)$$

By substituting the partial equilibrium points (0, 0), (0, 1), (1, 0), and (1, 1) into the expressions of the Jacobi matrix and the expressions of the trace, respectively, we can obtain the determinant det (J) of the Jacobi matrix and the value of the trace tr (J) of the game model when local governments assume the responsibility for the re-governance of pollution, which is shown in Table 3.

The positive and negative directions of $R_J - C_J$ and $R_J + G_B - G_A$ need to be further discussed according to Table 3, and the local stability of the different equilibrium points is discussed separately according to the different directional conditions.

**Table 3.** Determinant and trace values when local governments assume responsibility for renewed pollution governance.

| Equilibrium Points | det (J) | tr (J) |
|:---:|:---:|:---:|
| (0, 0) | $(R_J - C_J)(G_B - G_A)$ | $(R_J - C_J) + (G_B - G_A)$ |
| (1, 0) | $-(R_J - C_J)(R_J + G_B - G_A)$ | $-(R_J - C_J) + (R_J + G_B - G_A)$ |
| (0, 1) | $C_J(G_B - G_A)$ | $-C_J - (G_B - G_A)$ |
| (1, 1) | $-C_J(R_J + G_B - G_A)$ | $C_J - (R_J + G_B - G_A)$ |

From Table 4, the following conclusions can be drawn: the equilibrium results of the game between local governments and third-party governance institutions are (1, 0) and (0, 0). When $R_J - C_J > 0$, it means that if local governments regulate, the default compensation that can be obtained in the case of negative governance by third-party governance institutions is higher than the cost of regulation, so local governments will tend to adopt the "regulation" strategy. When $R_J + G_B - G_A < 0$, it means that if third-party governance institutions choose the "negative governance" strategy and are regulated by local governments, their default cost and negative governance costs will still be smaller than the active governance costs. At this time, third-party governance institutions are more likely to prefer the "negative governance" strategy, and the stable equilibrium point between the two parties is {active regulation, negative governance}. This shows that, at this time, although the benefits of local governments regulating third-party governance institutions are higher than the benefits of not regulating them, third-party governance institutions prefer default damages and eventually tend to choose the "negative governance" strategy due to the low cost of breach of contract. When $R_J - C_J < 0$, it means that if local governments regulate, they tend to adopt the "non-regulation" strategy if the liquidated damages they can obtain in the event of negative governance by third-party governance institutions are lower than the costs of regulation. Since the responsibilities of both parties cannot be defined when local governments do not regulate, third-party governance institutions are not responsible for the liquidated damages liability, so third-party governance institutions will tend to adopt the "negative governance" strategy as they will not receive liquidated damages, no matter how high the costs of default are set. This indicates that, at this time, the third-party governance contracts set up in the market are still relatively immature, local governments' regulatory costs are still high, and no additional benefits can be obtained from regulation.

**Table 4.** Local stability when local governments assume responsibility for renewed pollution governance.

| Scenario | Equilibrium Points | det (J) | tr (J) | Result |
|:---:|:---:|:---:|:---:|:---:|
| $R_J - C_J > 0$ and $R_J + G_B - G_A > 0$ | (0, 0) | − | Uncertain | Saddle |
| | (1, 0) | − | Uncertain | Saddle |
| | (0, 1) | − | Uncertain | Saddle |
| | (1, 1) | − | Uncertain | Saddle |
| $R_J - C_J > 0$ and $R_J + G_B - G_A < 0$ | (0, 0) | − | Uncertain | Saddle |
| | (1, 0) | + | − | ESS |
| | (0, 1) | − | Uncertain | Saddle |
| | (1, 1) | + | + | Unstable |
| $R_J - C_J < 0$ and $R_J + G_B - G_A > 0$ | (0, 0) | + | − | ESS |
| | (1, 0) | + | + | Unstable |
| | (0, 1) | − | Uncertain | Saddle |
| | (1, 1) | + | Uncertain | Unstable |
| $R_J - C_J < 0$ and $R_J + G_B - G_A < 0$ | (0, 0) | + | − | ESS |
| | (1, 0) | − | Uncertain | Saddle |
| | (0, 1) | − | Uncertain | Saddle |
| | (1, 1) | + | + | Unstable |

In this case, if local governments do not adopt the "regulation" strategy, third-party governance institutions can obtain more benefits by choosing the "negative governance" strategy. It can be seen that when the responsibility for poor governance is mainly borne by local governments, third-party governance institutions will tend to adopt the "negative governance" strategy, regardless of whether local governments adopt the "regulation" strategy. In this context, the third-party governance model cannot achieve effective pollution governance.

*3.2. The Evolutionary Stable Strategy for a Mixed Game When the Damaging Parties Are Responsible for the Re-Governance of Pollution*

There are various "failures" in local governments' regulation of third-party governance institutions [53]. If we want to further regulate the third-party pollution governance market, we need to readjust the boundary of responsibility between local governments and third-party governance institutions. Assuming that the existing policy stipulates that the damaging parties are mainly responsibility for pollution governance when there are issues with it, third-party governance institutions will be independently responsible for the problems caused by their negative governance, and will bear the cost of the re-governance of pollution, represented as $G_T$ (if local governments can effectively prove that third-party governance institutions have been ineffective in pollution governance, additional compensation to be paid by third-party governance institutions is denoted as $R_J$). At this point, the Jacobi matrix of the mixed game between the two sides is $J_2$:

$$J_2 = \begin{bmatrix} (1-2x)\left(-C_J + R_J - yR_J\right) & x(1-x)R_J \\ y(1-y)R_J & (1-2y)\left(xR_J + G_T + G_B - G_A\right) \end{bmatrix} \tag{14}$$

$$detJ_2 = (1-2x)\left(-C_J + R_J - yR_J\right)(1-2y)\left(xR_J + G_T + G_B - G_A\right) - x(1-x)R_J y(1-y)R_J \tag{15}$$

$$trJ_2 = (1-2x)\left(-C_J + R_J - yR_J\right) + (1-2y)\left(xR_J + G_T + G_B - G_A\right) \tag{16}$$

According to Table 5, the positive and negative directions of $G_T + G_B - G_A$, $R_J - C_J$, and $G_T + R_J + G_B - G_A$ need to be discussed, respectively, to determine the stability of each equilibrium point. When $G_T + G_B - G_A > 0$, $G_T + R_J + G_B - G_A > 0$, and $G_T + G_B - G_A < 0$, the cases of $G_T + R_J + G_B - G_A > 0$ and $G_T + R_J + G_B - G_A < 0$ need to be discussed separately, and the local stability cases generated by different parameter sizes are shown in Table 6a,b.

**Table 5.** When the damaging parties take the main responsibility for pollution governance, determinant value, and trace value.

| Equilibrium Points | det (*J*) | tr (*J*) |
|:---:|:---:|:---:|
| (0, 0) | $\left(-C_J + R_J\right)(G_T + G_B - G_A)$ | $\left(-C_J + R_J\right) + (G_T + G_B - G_A)$ |
| (1, 0) | $-\left(-C_J + R_J\right)\left(G_T + R_J + G_B - G_A\right)$ | $-\left(-C_J + R_J\right) + \left(G_T + R_J + G_B - G_A\right)$ |
| (0, 1) | $C_J(G_T + G_B - G_A)$ | $-C_J - (G_T + G_B - G_A)$ |
| (1, 1) | $-C_J\left(G_T + R_J + G_B - G_A\right)$ | $C_J - \left(G_T + R_J + G_B - G_A\right)$ |

It can be seen in Table 6a that when $G_T + G_B - G_A < 0$, the sum of the costs of re-governance and the original negative governance of third-party governance institutions is still smaller than the cost of active governance, and the evolutionary stable equilibrium points of both parties are {regulation, negative governance} and {no regulation, negative governance}. When $R_J > C_J$, the default compensation obtained by local governments when regulating is higher than the cost of regulation, and local governments will tend to choose the "regulation" strategy; when $R_J < C_J$, the default compensation obtained by local governments when regulating is lower than the costs of regulation, and local governments will tend to adopt the "non-regulation" strategy. At this point, regardless of whether local governments choose the "regulation" strategy, the benefits of third-party governance

institutions choosing the "negative governance" strategy are higher than the benefits of active governance, and third-party governance institutions tend to adopt the "negative governance" strategy. This means that even if third-party governance institutions assume the responsibility for re-governance, if the costs of re-governance are low due to a lack of strict control over re-governance standards, third-party governance institutions seeking to maximize their profits will have no incentive to provide active governance, resulting in poor outcomes for third-party institutions and a failure to achieve the government's goal of guiding and promoting the third-party governance market. As can be seen in Table 6b, the sum of the costs of re-governance and the original negative governance by third-party governance institutions is higher than the cost of active governance, regardless of whether local governments choose to regulate or not, and the evolutionary stability equilibrium between the two parties is {regulation, active governance} and {non-regulation, active governance}. When $G_T + G_B - G_A > 0$ and $G_T + R_J + G_B - G_A > 0$, at this time, if the net benefits of negative governance (the sum of the costs of negative governance, the costs of re-governance, and the costs of default) are higher than the net benefits of active governance for third-party governance institutions, regardless of whether local governments choose the "regulation" strategy, third-party governance institutions will tend to adopt the "active governance" strategy. Regardless of whether the benefits of defaulting from regulation are higher than the costs of regulation, when third-party governance institutions choose the "active governance" strategy, local governments will tend to adopt the "non-regulation" strategy, and the evolutionary stability strategy for both parties will be (0, 1).

**Table 6.** (**a**) Local stability case when $G_T + G_B - G_A < 0$. (**b**) Local stability case when $G_T + G_B - G_A > 0$.

| (a) | | | | |
|---|---|---|---|---|
| **Scenario** | **Equilibrium Points** | **det (*J*)** | **tr (*J*)** | **Result** |
| $R_J > C_J$ and $G_T + R_J + G_B - G_A > 0$ | (0, 0) | − | Uncertain | Saddle |
|  | (1, 0) | − | Uncertain | Saddle |
|  | (0, 1) | − | Uncertain | Saddle |
|  | (1, 1) | − | Uncertain | Saddle |
| $R_J < C_J$ and $G_T + R_J + G_B - G_A > 0$ | (0, 0) | + | − | ESS |
|  | (1, 0) | + | + | Unstable |
|  | (0, 1) | − | Uncertain | Saddle |
|  | (1, 1) | − | Uncertain | Saddle |
| $R_J > C_J$ and $G_T + R_J + G_B - G_A < 0$ | (0, 0) | − | Uncertain | Saddle |
|  | (1, 0) | + | − | ESS |
|  | (0, 1) | − | Uncertain | Saddle |
|  | (1, 1) | + | + | Unstable |
| $R_J < C_J$ and $G_T + R_J + G_B - G_A < 0$ | (0, 0) | + | − | ESS |
|  | (1, 0) | − | Uncertain | Saddle |
|  | (0, 1) | − | Uncertain | Saddle |
|  | (1, 1) | + | + | Unstable |
| (b) | | | | |
| **Scenario** | **Equilibrium Points** | **det (*J*)** | **tr (*J*)** | **Result** |
| $R_J > C_J$ and $G_T + G_B - G_A > 0$ | (0, 0) | + | + | Unstable |
|  | (1, 0) | − | Uncertain | Saddle |
|  | (0, 1) | + | − | ESS |
|  | (1, 1) | − | Uncertain | Saddle |
| $R_J < C_J$ and $G_T + G_B - G_A > 0$ | (0, 0) | − | Uncertain | Saddle |
|  | (1, 0) | + | + | Unstable |
|  | (0, 1) | + | − | ESS |
|  | (1, 1) | − | Uncertain | Saddle |

## 4. Simulation Analysis

To further verify the dynamic evolutionary path of the system under different scenarios for local governments and third-party governance institutions, MATLAB R2022a software was used to simulate the dynamic evolutionary process of strategy selection between the two parties under different parameter situations and different initial states to verify the accuracy of the model results.

If $R_J = 4$, $C_J = 3$, $G_B = 5$, and $G_A = 10$, at this time, the compensation benefits $R_J$ received by local governments when regulating third-party governance institutions for negative pollution governance are higher than the costs of regulation $C_J$; the costs of default for third-party governance institutions and the costs of negative governance $(R_J + G_B)$ are smaller than the costs of active governance $G_A$; and the requirements that $R_J - C_J > 0$ and $R_J + G_B - G_A < 0$ are obtained to satisfy the condition. The simulation results of the system evolutionary trend are shown in Figure 1a. At this time, local governments tended to adopt the "regulation" strategy and third-party governance institutions tended to adopt the "negative governance" strategy, and the stable equilibrium point of both parties was $(1, 0)$.

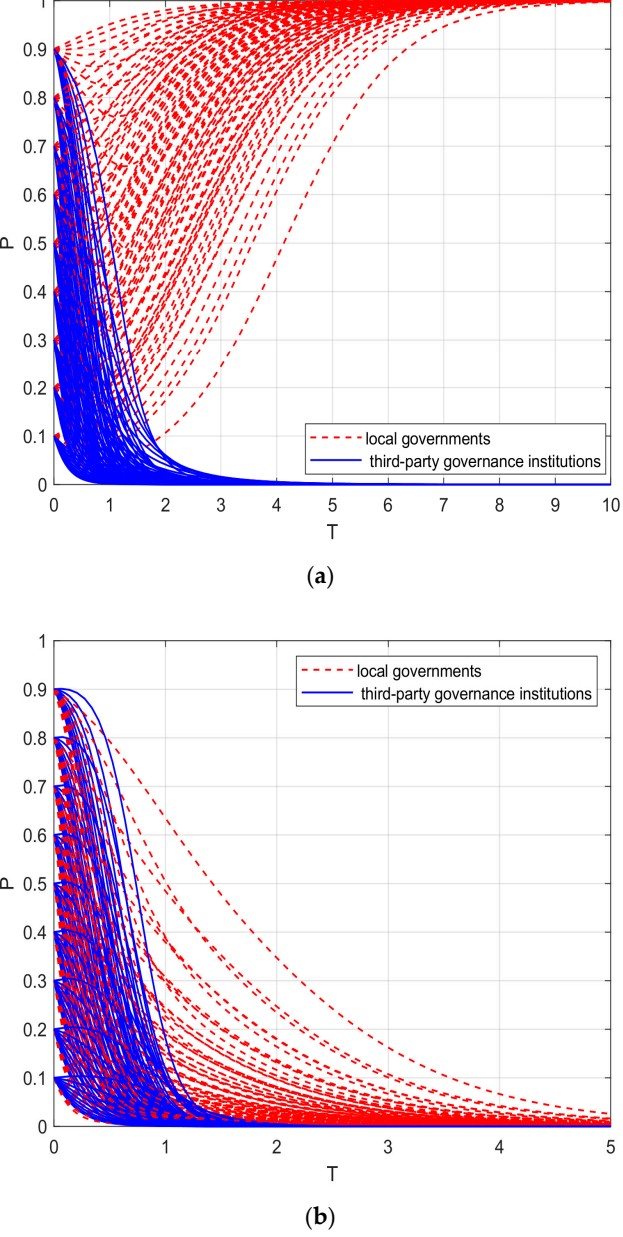

(a)

(b)

**Figure 1.** *Cont.*

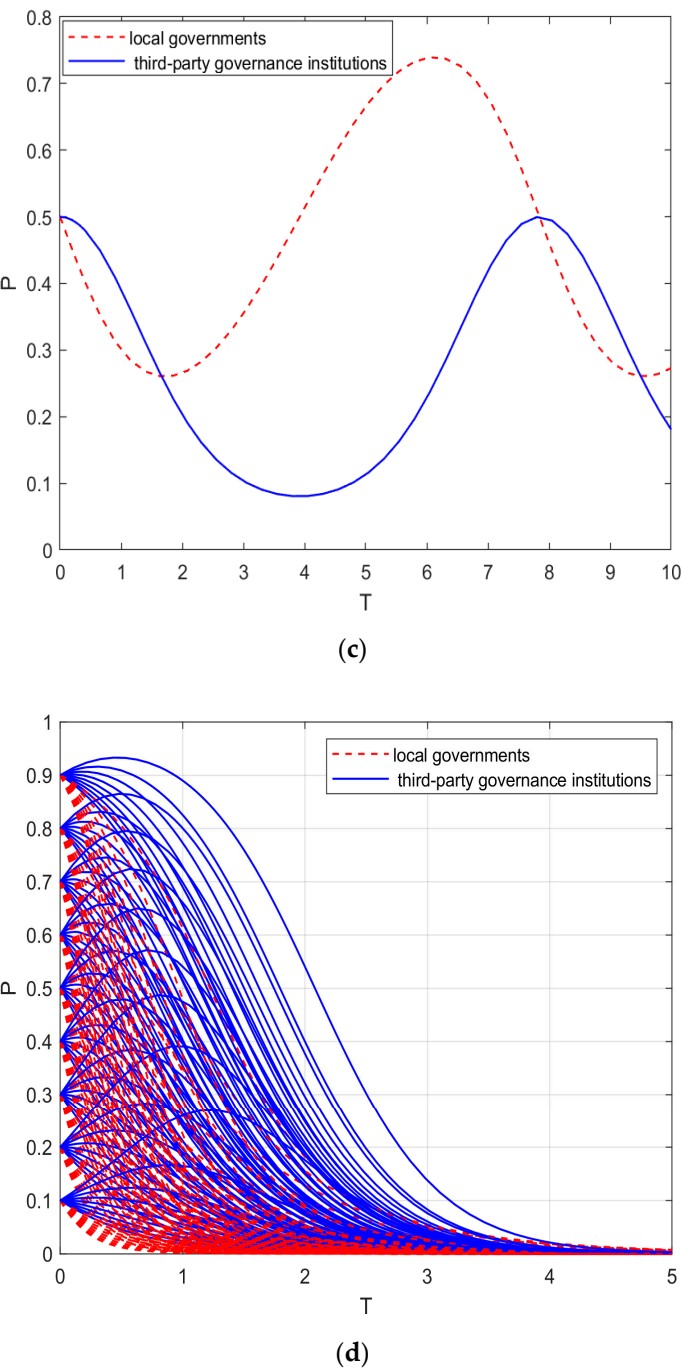

**Figure 1.** *Cont.*

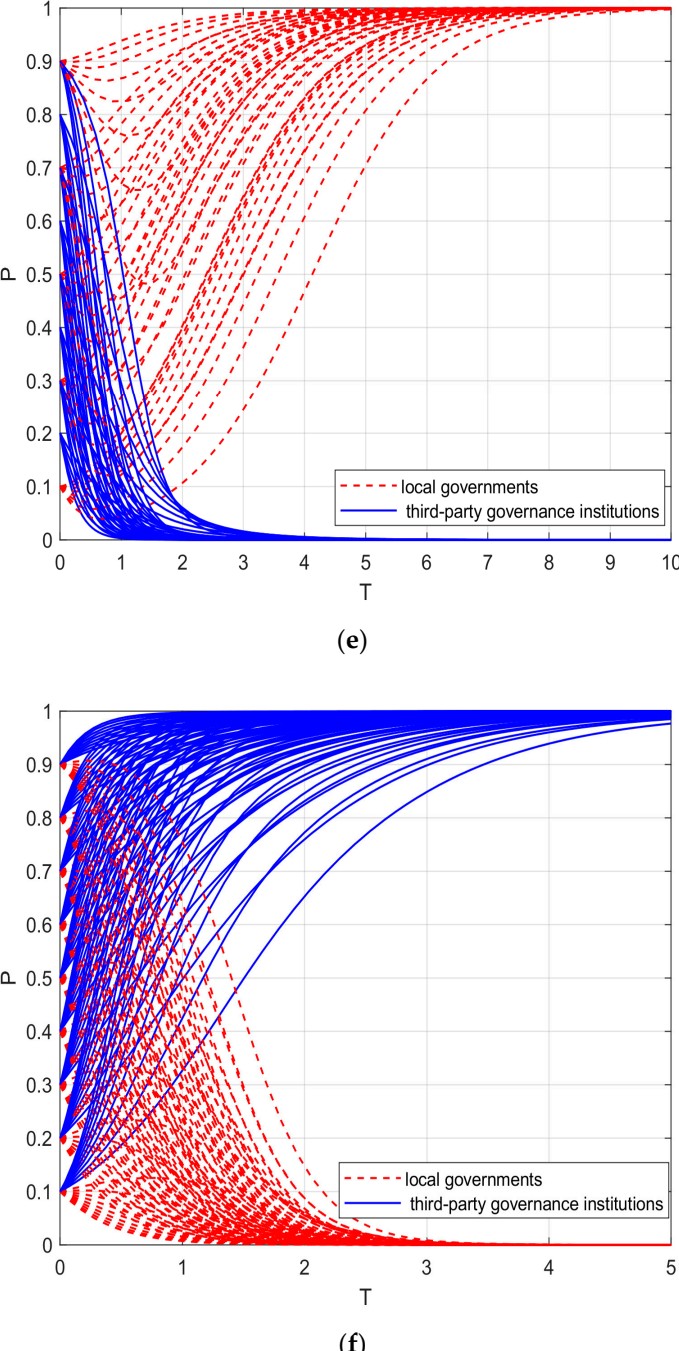

**Figure 1.** Evolutionary stabilization strategies for both parties when local governments assume responsibility for renewed pollution control. Evolutionary stabilization strategies for both parties when the damaging party assumes the main responsibility for pollution control. (**a**) The system evolution simulation result when $R_J - C_J > 0$ and $R_J + G_B - G_A < 0$. (**b**) The system evolution simulation result when $R_J - C_J < 0$. (**c**) The system evolution simulation result when $R_J > C_J$, $R_J + G_B - G_A < 0$, and $G_T + R_J + G_B - G_A > 0$. (**d**) The system evolution simulation result when $R_J < C_J$, $R_J + G_B - G_A < 0$, and $G_T + R_J + G_B - G_A > 0$. (**e**) The system evolution simulation result when $R_J < C_J$ and $G_T + R_J + G_B - G_A > 0$. (**f**) The system evolution simulation result when $R_J + G_B - G_A > 0$.

If $R_J = 2$, $C_J = 3$, $G_B = 4$, and $G_A = 10$, or $R_J = 7$, $C_J = 8$, $G_B = 4$, and $G_A = 10$, then the compensation benefits $R_J$ received by local governments when regulating third-party governance institutions' negative governance are lower than the costs of regulation $C_J$. At this time, regardless of whether the costs of third-party governance institutions' default and the costs of negative governance $(R_J + G_B)$ are higher than the costs of active governance, local governments are more likely to choose the "non-regulation" strategy, and third-party governance institutions will tend to adopt the "negative governance" strategy. The simulation results of the evolutionary trend of the system are shown in Figure 1b). The simulation results showed that when local governments assumed responsibility for ineffective pollution governance, third-party governance institutions tended to adopt the "non-regulation" strategy, regardless of whether local governments regulated, and the third-party governance market could not effectively play the role of the governance market at this time.

Let $R_J = 4$, $C_J = 3$, $G_B = 5$, $G_A = 10$, and $G_T = 3$. At this time, the compensation benefits $R_J$ received by local governments when regulating third-party governance institutions for negative governance are higher than the costs of regulation $C_J$; the costs of default by third-party governance institutions and the costs of negative governance $(R_J + G_B)$ are smaller than the costs of active governance $G_A$; the costs of default by third-party governance institutions, the costs of negative governance, and the costs of re-governance $(R_J + G_B + G_T)$ are greater than the costs of active governance $G_A$; and the conditions that $R_J > C_J$, $R_J + G_B - G_A < 0$, and $G_T + R_J + G_B - G_A > 0$ are satisfied. At this point, the evolutionary trend of both parties is in a cyclical circular motion, i.e., when local governments regulate, third-party governance institutions will tend to adopt the "active governance" strategy; when third-party governance institutions are inclined to choose "active governance", local governments will tend to adopt the "non-regulation" strategy; and when local governments choose not to regulate, third-party governance institutions will tend to adopt the "negative governance" strategy. Therefore, both parties cannot be in a stable equilibrium state. The system evolution simulation results are shown in Figure 1c.

Let $R_J = 4$, $C_J = 5$, $G_B = 5$, $G_A = 10$, and $G_T = 3$. At this time, the compensation benefits $R_J$ received by local governments when regulating third-party governance institutions for negative governance are lower than the costs of regulation $C_J$; the costs of default by third-party governance institutions and the costs of negative governance $(R_J + G_B)$ are smaller than the costs of active governance $G_A$; the sum of the costs of the default by third-party governance institutions, the costs of negative governance, and the costs of re-governance $(R_J + G_B + G_T)$ is greater than the cost of active governance $G_A$; and the conditions that $R_J < C_J$, $R_J + G_B - G_A < 0$, and $G_T + R_J + G_B - G_A > 0$ are satisfied. At this point, local governments will tend to choose the "non-regulation" strategy and third-party governance institutions will tend to adopt the "negative governance" strategy. The system evolution simulation results are shown in Figure 1d.

Let $R_J = 4$, $C_J = 3$, $G_B = 2$, $G_A = 10$, and $G_T = 3$. At this time, the compensation benefits $R_J$ obtained by local governments when regulating third-party governance institutions for negative governance are higher than the costs of regulation $C_J$; the sum of the costs of default by third-party governance institutions, the costs of negative governance, and the costs of re-governance $(R_J + G_B + G_T)$ is greater than the costs of active governance; and we obtain $R_J < C_J$ and $G_T + R_J + G_B - G_A > 0$. At this time, local governments will be more likely to choose the "regulation" strategy and third-party governance institutions will tend to choose the "active governance" strategy. The system evolution simulation results are shown in Figure 1e.

Let $R_J = 4$, $C_J = 5$, $G_B = 8$, $G_A = 10$, and $G_T = 3$ or let $R_J = 4$, $C_J = 3$, $G_B = 8$, $G_A = 10$, and $G_T = 3$. At this time, the costs of negative governance and re-governance by third-party governance institutions are higher than the costs of active governance, so, regardless of whether the compensation benefits $R_J$ received by local governments when regulating third-party governance institutions for negative governance are higher than the costs of regulation $C_J$, third-party governance institutions will be inclined to choose

the "active governance" strategy. At this time, no regulation is the optimal evolutionary strategy of third-party governance institutions. The system evolutionary simulation results are shown in Figure 1f.

## 5. Discussion

(1) Regardless of whether local governments assume the responsibility for re-governance or whether the damaging parties assume the responsibility for the main body of governance, if local government regulation can obtain liquidated damage benefits that are higher than the costs of regulation, and the total costs of the negative governance by third-party governance institutions are higher than the costs of active governance, then the local government regulation of third-party governance institutions cannot enable third-party governance institutions to exist in a long-term active governance state. This means that only when local governments adopt the "regulation" strategy will third-party governance institutions adopt the "active governance" strategy. Once local governments cease regulatory oversight, the third-party governance market will tend to revert to the "negative governance" strategy [54].

(2) If local governments assume the main responsibility for governance, third-party governance institutions tend to choose the "negative governance" strategy, regardless of whether local governments regulate it or not, and the third-party governance market cannot effectively play the role of the governance market in this context. Only when local governments adopt the "regulation" strategy will third-party governance institutions adopt the "active governance" strategy. This reveals that even though third-party governance has achieved the separation of the "pollution generation" and "pollution control" entities, it would be difficult to achieve effective pollution governance in rural areas if the responsibility allocation issues continue to be addressed under the "dual governance" model.

(3) If third-party governance institutions assume the main responsibility for governance, and the total costs of the re-governance (costs of re-governance, costs of liquidated damages, and costs of active governance) caused by negative governance are higher than the total costs of active governance, even if local governments adopt the "non-regulation" strategy, third-party governance institutions will still adopt the "active governance" strategy, realizing the effective development of the third-party governance model. If the standard of re-governance is not strictly controlled, resulting in low costs of re-governance, and the total costs of the negative governance by third-party governance institutions are less than the costs of active governance, then even if the third-party governance institutions assume responsibility for the governance, there will be a situation where third-party governance institutions pursuing maximum benefits will tend to choose the "negative governance" strategy, regardless of whether local governments choose to regulate. This will not achieve the purpose of government guidance, which is to promote the third-party governance market.

## 6. Conclusions and Recommendations

### 6.1. Conclusions

In the context of many issues, such as the imperfect supervision and audit mechanisms of the third-party governance market, unclear standards of responsibility sharing, and the inconsistent operation of judicial practice, we attempted to explore the path of effective third-party governance by constructing a game model between local governments and third-party governance institutions, analyzing the behavioral strategies of local governments and third-party governance institutions, and analyzing the logic and ideas behind the behavioral choices of each subject. Through the evolutionary game, the following conclusions were reached:

(1) Third-party rural environmental governance places higher demands on the government's capacity for governance. For instance, the establishment of boundaries for

re-governance responsibility in the third-party governance of rural environmental pollution is of the utmost importance.

(2) When local governments bear the primary responsibility for governance, regardless of whether they provide regulatory oversight, third-party governance institutions will tend to adopt a passive approach. In such cases, the third-party governance market will fail to effectively fulfill its role in governance.

(3) By reconstructing the third-party governance market model and dividing the main responsibility for pollution governance among the damaging parties, it is possible to achieve active governance by third-party governance institutions without the need for regulation by local governments.

*6.2. Recommendations*

The conclusions in this paper have significant policy implications for further improving the market for the third-party governance of rural environmental pollution.

First, the detailed responsibilities and obligations of third-party governance service contracts for rural environments should be developed as much as possible. As a new model of rural environmental pollution governance, third-party pollution governance has achieved a separation of the main bodies of "pollution production" and "pollution governance", and, if we continue to solve the responsibility allocation problem according to the "dual governance" model, it will be difficult to achieve the effective management of rural environmental pollution. Therefore, the previous model of local governments taking responsibility for re-governance under a dual structure should be re-designed to have the damaging parties take responsibility for re-governance, i.e., if the negative governance caused by third-party governance institutions is unsatisfactory, third-party governance institutions should bear the responsibility.

Second, the regulatory costs of local governments should be reduced as much as possible. For inconsistent third-party governance institutions, local governments can optimize the access system of the third-party governance market for rural environments, and exclude from the market those institutions with repeated negative governance performances and poor credit qualifications to reduce the difficulty for local governments in selecting third-party governance institutions.

Finally, incentives and penalties should be further standardized. For rural areas with better environmental governance, financial support or preferential rural financing policies should be given to fundamentally motivate local governments to participate in third-party governance. Penalties should be increased for third-party governance institutions that choose negative governance strategies, and pollution re-governance standards should be improved to ensure that the total costs of default by third-party governance institutions' negative governance are higher than the costs of active governance, thus reducing the possibility of negative governance by third-party governance institutions.

*6.3. Limitations and Research Prospects*

Several limitations exist in our work. Due to the relative novelty of implementing third-party environmental pollution control in rural areas in China, there was a lack of specific cases and data for detailed analysis. As a result, the analysis presented in this paper primarily remains at the theoretical level. In terms of model construction, only the game behavior between local governments and third-party governance institutions was considered, without analyzing the impact after incorporating other stakeholders.

In future, we may include a significant stakeholder group, the villagers, in the model to conduct a three-party evolutionary game analysis. On this basis, we will analyze cases and data from real-world implementations of third-party environmental governance to provide effective policy recommendations for developing more accurate and efficient third-party governance solutions.

**Author Contributions:** Conceptualization, Q.W. (Qianwen Wu); methodology, Y.D.; software, Q.W. (Qiangqiang Wang); Formal analysis, Q.W. (Qianwen Wu); Writing—original draft, Q.W. (Qianwen Wu); Writing—review & editing, Y.D.; Supervision, Y.D. All authors have read and agreed to the published version of the manuscript.

**Funding:** This study was supported by the National Natural Science Foundation of China (71973027).

**Institutional Review Board Statement:** This study has no ethical implications and therefore does not require ethical approval.

**Informed Consent Statement:** Informed consent was obtained from all subjects involved in the study.

**Data Availability Statement:** This study did not report data.

**Conflicts of Interest:** The authors declare no conflict of interest.

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
