# Peer review of "Analysis of Strategy Selection in Third-Party Governance of Rural Environmental Pollution"

_sustainability, doi:10.3390/su15118746_

Round 1
Reviewer 1 Report
Review for the submitted paper entitled “Analysis of the Strategy Selection of Third-Party Governance of Rural Environmental Pollution”:
The authors analyzed different evolutionary game strategies undertaken by local governments and third-party governance institutions under different institutional designs assuming responsibility for retreatment and damaging parties assuming responsibility for re-governance.
This topic is very relevant and suitable for the sustainability journal. Overall, the paper merits publication, but with major revision.
The paper needs to be proofread by a native speaker to vastly improve its readability.
The authors should seek to include literature from a wider geographical scope and context within the introductory paragraphs and also the discussion. More references should be added.
Section Conclusions and Suggestions should be changed to Conclusions and Recommendations.
Please include a few policy implications of your results.
Reviewer 2 Report
The topic of the manuscript is relevant and can be published!
comments:
1. I recommend expanding the review of the literature, because today the manuscript looks almost monocontinental! There are many scientific and design materials in other (modern) countries!
2. The drawings need to be technically refined and the technical execution changed (according to the publishing house's requirements)! And also the grammar of the text (spaces, periods, etc., for example, line 140, 146)
3. The "discussion" section - there are general rules for writing such sections, I recommend reworking them!
Reviewer 3 Report
Dear Authors,
thank you for interesting and valuable research. However, I have a few suggestions to improve your paper.
1. I do feel that Introduction part could be better based on scientific literature, e.g. in line 88 you write some scholars... but there is only one reference to that and there are more examples of this.
2. Please elaborate the part on evolution game theory. This theory is well described and developed but you do not reveal the complexity and richness of the theory.
3. Is the Hypothesis 1 really a hypothesis? I truly believe that it is a constant but not a hypothesis. Please elaborate and ground this hypothesis better.
4. How are your hypotheses related to scientific literature? There should be explanation here or at least at your results/conclusions part.
5.How does the result of your research contributes to scholar discussion and to the field? Please refer your results to scientific discussion and not only to policy recommendations.
6. What are the limitations of your research? What is your future research in this field? What should be done and why?
Language is good. Please review introduction as there are some typo; the rest of the text is of a good quality regarding English language.
Reviewer 4 Report
The manuscript presents some modeling and simulations for third party governance of rural environmental pollution.
The English needs a lot of improvement in terms of grammar and some extremely long sentences, which make it hard to read.
The abstract has no clear objective.
Keywords are repeated from the title.
There is no citation for the document mentioned on Line 37.
Line 100 mentions “other scholars” but only has 1 citation.
Line 121 presents an objective that does not mention models and simulations, which is misleading.
Hypothesis 1 is not falsifiable. It can only be posited.
Hypotheses 2 and 3 must be made clearer. ¿What is it that the manuscript is trying to prove?
Hypotheses should be clearly accepted or rejected.
Do results begin on Line 198? This should be made clear.
Page 7: tables should be mentioned in the text before they appear.
Table 4: What does “saddle” and ESS mean?
Discussion section is really a conclusion, as Line 439 clearly states.
The manuscript needs a proper discussion, where results are compared to previous research. There are many references cited that can be used for this purpose.
Line 482: this should be a recommendations section.
The grammar needs a lot of work. There are very long sentences that make the manuscript hard to read.
Round 2
Reviewer 1 Report
Thanks for submitting and revising this manuscript. It is an interesting paper and the insights have high value for Sustainability. I would recommend accepting for publication.
Reviewer 2 Report
Recommend for publication!
Reviewer 4 Report
The authors did a thorough job of addressing the main concerns.